# On the expressivity of bi-Lipschitz normalizing flows

**Alexandre Vérine** [1]   **Benjamin Negrevergne** [1]   **Fabrice Rossi** [2]   **Yann Chevaleyre** [1]

## Abstract

An invertible function is *bi-Lipschitz* if both the function and its inverse have bounded Lipschitz constants. Nowadays, most Normalizing Flows are bi-Lipschitz by design or by training to limit numerical errors (among other things). In this paper, we discuss the expressivity of bi-Lipschitz Normalizing Flows and identify several target distributions that are difficult to approximate using such models. Then, we characterize the expressivity of bi-Lipschitz Normalizing Flows by giving several lower bounds on the Total Variation distance between these particularly unfavorable distributions and their best possible approximation. Finally, we discuss potential remedies which include using more complex latent distributions.

## 1. Introduction

A number of recent publications have demonstrated the benefits of constructing machine learning models with a small Lipschitz constant. First, models with a small Lipschitz constant have been linked with better generalization capabilities, both in terms of true risk (Bartlett et al., 2017), and adversarial risk (Farnia et al., 2018). In addition, models with a small Lipschitz constraint are more stable during training, and are less prone to numerical errors, a property which is particularly important in the context of invertible neural networks and normalizing flows (Behrmann et al., 2021).

Unfortunately, enforcing a small Lipschitz constant, either by design, or using regularization during training, can impede the ability of a model to fit the data distribution. Based on this observation, several researchers have studied the limitations of neural networks with bounded Lipschitz con-

[1]Université Paris-Dauphine, PSL Research University, CNRS, LAMSADE, Paris, France [2]Université Paris-Dauphine, PSL Research University, CNRS, CEREMADE, Paris, France. Correspondence to: Alexandre Vérine <alexandre.verine@dauphine.psl.eu>.

Third workshop on *Invertible Neural Networks, Normalizing Flows, and Explicit Likelihood Models* (ICML 2021). Copyright 2021 by the author(s).

stant. In particular Tanielian et al. (2020) was able identify a family of target distributions with disconnected support that cannot be fitted with a GAN with a bounded Lipschitz constant.

In this paper we focus on the impact of the Lipschitz constraints on normalizing flows. Normalizing flows are often not only Lipschitz, but *bi-Lipschitz*, meaning that both the mapping function and its inverse have bounded Lipschitz constant. For example, Additive Coupling, neural ODE and Residual Networks are bi-Lipschitz by design. Other types of normalizing flows, can also be trained to be bi-Lipschitz, in order to avoid exploding inverses (Behrmann et al., 2021). We study the expressivity of normalizing flows with bounded Lipschitz constant and discuss the impact of the bi-Lipschitz constant on the Total Tariation distance. More precisely, we give several lower bounds on the total variation distance between the generated distribution and the target distribution, in some (particularly unfavorable) training settings.

## 2. Background

A *normalizing flow* is an invertible density model in which both density estimation and sampling can be done efficiently. In short, training a normalizing flow consists in learning an invertible mapping between a data space $\mathcal{X}$ and a latent space $\mathcal{Z}$. Typically, the forward direction $F : \mathcal{X} \to \mathcal{Z}$ (i.e. the *normalizing* direction) is tractable and exact and the inverse direction $F^{-1} : \mathcal{Z} \to \mathcal{X}$ (i.e. the *generative* direction) either has a closed form, or can be approximated using an iterative algorithm.

Suppose that $P^*$ is the true data distribution over $\mathcal{X}$, and that $P^*$ admits a density function denoted $p^*$ that we wish to approximate. We first chose a $d$-dimensional Gaussian distribution $Q$ over $\mathcal{Z}$ (a.k.a. the latent space), and its density function $q(\boldsymbol{z}) = \frac{1}{(\sqrt{2\pi})^d} e^{-\frac{1}{2}\|\boldsymbol{z}\|_2^2}$. Then, we can define $\hat{p}$, the approximation of $p^*$, based on $q$ and the mapping $F : \mathcal{X} \to \mathcal{Z}$, using a simple change of variable formula:

$$\forall \boldsymbol{x} \in \mathcal{X}, \quad \hat{p}(\boldsymbol{x}) = |\det \mathrm{Jac}_F(\boldsymbol{x})| \, q(F(\boldsymbol{x})) \qquad (1)$$

Note the estimated probability $\widehat{P}(A)$ of any event $A \subseteq \mathcal{X}$

can be retrieved as follows:

$$\widehat{P}(A) = Q(F(A)) = \int_{F(A)} q(\boldsymbol{z})d\boldsymbol{z}$$

As seen in Equation 1, performing density estimates requires computing the determinant of the Jacobian matrix which can be large in practice, thus most normalizing flows have been specifically designed to make this computation efficient.

## 2.1. Bi-Lipschitz Normalizing Flows

In this paper, we focus on bi-Lipschitz normalizing flows, which is a mapping $F$ whose Lipschitz constants are bounded in both directions. More specifically, we define the bi-Lipschitz property as follows.

**Definition 2.1.** *A bijective function $F : \mathcal{X} \subset \mathbb{R}^d \to \mathcal{Z} \subset \mathbb{R}^d$ is said to be $(L_1, L_2)$-bi-Lipschitz if $F$ is $L_1$-Lipschitz and its inverse $F^{-1}$ is $L_2$-Lipschitz, i.e.:*

$$\forall \boldsymbol{x}_1, \boldsymbol{x}_2 \in \mathcal{X}, \quad \|F(\boldsymbol{x}_1) - F(\boldsymbol{x}_2)\| \leq L_1 \|\boldsymbol{x}_1 - \boldsymbol{x}_2\|$$

*and*

$$\forall \boldsymbol{z}_1, \boldsymbol{z}_2 \in \mathcal{Z}, \quad \|F^{-1}(\boldsymbol{z}_1) - F^{-1}(\boldsymbol{z}_2)\| \leq L_2 \|\boldsymbol{z}_1 - \boldsymbol{z}_2\|$$

Alternatively, since the mapping $F$ is bijective, the bi-Lipschitz continuity can be expressed over $F$ only as:

$$\frac{1}{L_2} \|\boldsymbol{x}_1 - \boldsymbol{x}_2\| \leq \|F(\boldsymbol{x}_1) - F(\boldsymbol{x}_2)\| \leq L_1 \|\boldsymbol{x}_1 - \boldsymbol{x}_2\|$$

However, enforcing the bi-Lipschitz continuity of $F$ results in a bounded determinant for the Jacobian matrix:

**Proposition 2.1.** $\mathrm{Jac}_F$ *satisfies for all $\boldsymbol{x} \in \mathcal{X}$:*

$$\frac{1}{L_2^d} \leq |\det \mathrm{Jac}_F(\boldsymbol{x})| \leq L_1^d$$

As we will show in the rest of this paper, this can limit the expressivity of normalizing flows.

This is relevant, because many normalizing flows are bi-Lipschitz in practice, for example, the i-ResNet (Behrmann et al., 2019) and the Residual Flow (Chen et al., 2020) are both based on residual atomic blocks $f_i = I_d + g_i$. Their invertibility is ensured by the Lipschitz constant $\mathrm{Lip}(g_i) \leq L < 1$. If $F$ is composed of $m$ residual blocks such that $F = f_m \circ \cdots \circ f_1$, then the overall bi-Lipschitz constants satisfy $\mathrm{Lip}(F) \leq (1 + L)^m$ and $\mathrm{Lip}(F^{-1}) \leq 1/(1 - L)^m$. Alternatively, in Glow (Kingma & Dhariwal, 2018) with atomic blocks $W_i = P_i L_i (U_i + \mathrm{diag}(s_i))$, the bi-Lipschitz constants statisfy: $\mathrm{Lip}(F) \leq \prod_i^m \|W_i\|_2$ and $\mathrm{Lip}(F^{-1}) \leq \prod_i^m \|W_i^{-1}\|_2$. Consequently, the bi-Lipschitzness constraints on either the function or its Jacobian determinant can be released by increasing the depth of the network but, by doing so, the stability of the inverse can be affected (Behrmann et al., 2021).

## 2.2. Assessing the learning abilities

Our goal is to understand how the bi-Lipschitz property affects the approximation ability of the network. To do so, we will compare the true data distribution $P^*$ and its density $p^*$ with the learned distribution $\widehat{P}$ and its density $\hat{p}$.

To evaluate how the true distribution $P^*$ and the generated distribution $\widehat{P}$ differ, we use the Total Variation (TV) distance defined as:

$$\mathcal{D}_{\mathrm{TV}}(P^*, \widehat{P}) = \sup_A |P^*(A) - \widehat{P}(A)|$$

# 3. Lower Bounds on the TV Distance

## 3.1. A bound on bi-Lipschitz normalizing flow for any subset $A$

The first theorem is a lower bound on the TV distance between the learned distribution and the target distribution in a general setting. Intuitively, the idea is to find an arbitrary subset $A$ that is sufficiently concentrated so that the Lipschitz constrained mapping can not concentrate enough weight form the Gaussian distribution onto this subset.

**Theorem 3.1** (bi-Lipschitz mappings fail to capture high density subset). *Let $F$ be $(L_1, L_2)$-bi-Lipschitz and $\eta_A = \frac{P^*(A)}{\mathrm{vol}(A)}$ be the average density over any subset $A \subset \mathbb{R}^d$. Then:*

$$\mathcal{D}_{\mathrm{TV}}(P^*, \widehat{P}) \geq \sup_A \mathrm{vol}(A) \left( \eta_A - \left( \frac{L_1}{4 L_2 \sqrt{2\pi}} \right)^d \right)$$

*Therefore, if there is a subset $A$ that satisfies $\eta_A > \left( \frac{L_1}{4 L_2 \sqrt{2\pi}} \right)^d$, then the TV is necessarily strictly positive.*

The proof of this Theorem is given in appendix A.1.

Remark that the bound in Theorem 3.1 depends on both Lipschitz constraints $L_1$ and $L_2$. If a subset $A$ is found to be very dense, the mapping will not be able to expand the given volume of $A$ to match the lower density of the Gaussian density because of $L_1$. On the other hand, the point with the highest density within $A$ will be matched with the highest point on the Gaussian density but all its neighbourhood has to me moved by a factor of $1/L_2$. The main advantage of this formulation is to apply to any subset of the data space, but at the expense of a loose bound on the TV.

## 3.2. Bounds for specific subset $B_{R,\boldsymbol{x}}$

The bound in Theorem 3.1 can be further improved by making assumptions on the structure of the subset $A$. We choose to focus on $l_2$ balls instead of arbitrary subsets.

Let $B_{R,\boldsymbol{x}_0}$ be the $l_2$ ball with center $\boldsymbol{x}_0$ and radius $R$ (i.e. $B_{R,\boldsymbol{x}_0} = \{\boldsymbol{x} \in \mathcal{X} : \|\boldsymbol{x} - \boldsymbol{x}_0\|_2 \leq R\}$). Then we can

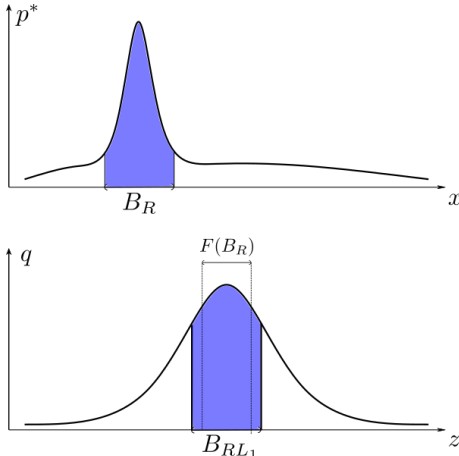

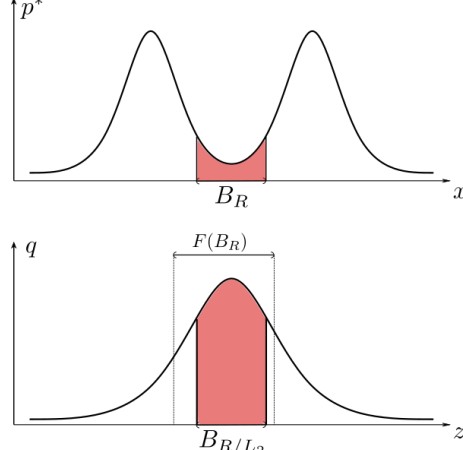

*Figure 1.* Example of a target distribution where theorem 3.2 applies: the subset $B_R$ concentrates most of the weight in $P^*(B_R)$, but $\widehat{P}(B_R) = Q(F(B_R))$ can only be as large as $Q(B_{R_{L_1}})$.

*Figure 2.* Example of a target distribution for which Theorem 3.3 applies: the subset $B_R$ concentrates little weight in $P^*(B_R)$, but $\widehat{P}(B_R) = Q(F(B_R))$ can only be as small as $Q(B_{R/L_2})$.

show that both high density balls and low density ones are difficult to fit properly, the former because of the Lipschitz constraint of $F$, the latter because of the Lipschitz constraint of $F^{-1}$. We first consider high density balls.

**Theorem 3.2** (NF with a $L_1$-Lipschitz mapping $F$ fails to capture high density balls). *Let $F$ be $L_1$-Lipschitz. Then:*

$$\mathcal{D}_{\mathrm{TV}}(P^*, \widehat{P}) \geq \sup_{R, \boldsymbol{x}_0} \left( P^*(B_{R, \boldsymbol{x}_0}) - \frac{RL_1}{\sqrt{\pi}} \right)$$

$$\mathcal{D}_{\mathrm{TV}}(P^*, \widehat{P}) > \sup_{R, \boldsymbol{x}_0} \left( P^*(B_{R, \boldsymbol{x}_0}) - 4d^{1/4}RL_1 \right)$$

*Therefore, if we find a ball for which the true measure satisfies $\frac{P^*(B_{R, \boldsymbol{x}_0})}{R} > \frac{L_1}{\sqrt{\pi}}$ or $\frac{P^*(B_{R, \boldsymbol{x}_0})}{R} > 4d^{1/4}L_1$, then the TV is necessarily strictly positive.*

The theorem 3.2 highlights the effect of the Lipschitz constraint of the forward mapping $F$. If a ball has a high probability in the data space $P^*(B_R)$, then the probability assigned to this ball is at most $Q(B_{RL_1})$ in the latent space and is upper bounded by $RL_1/\sqrt{\pi}$ (Ball, 1993). A one dimensional representation of a pathological case for theorem 3.2 is shown on figure 1. In other words no ball with a high enough density in data space can be expended sufficiently to have a matching probability in the latent space. Note that we could use a closed-form of $Q(B_{RL_1})$ but it is less open to interpretation than the approximations we have made.

Conversely, the mapping being bi-Lipschitz, the mapping can not contract arbitrarily. If there is a low density zone mapped on the maximum of the Gaussian density, then the Normalizing Flow cannot reduce enough the probability of the corresponding zone. Notice that the assumption of a low density zone is strong but fairly reasonable. For instance,

one can observe a multi-modal density with fairly well separated modes. If the modes are roughly equiprobable, we expect a mapping to assign those modes in balanced way around the mode of the Gaussian distribution in the latent space. Therefore, the low density ball is mapped on a zone wider than the ball $B_{R/L_2}$ and consequently the Gaussian measure associated is lower bounded by $Q(B_{R/L_2})$ as illustrated on the one dimensional example on figure 2. Despite the lower bounds established by (Pinelis, 2020), there is no reasonably interpretable bounds, therefore we use the closed-form that is expressed with the Gamma function $\Gamma$ and the incomplete gamma function $\gamma$. The numerical approximations of the closed form are given in figure 3. We can observe that the higher the dimension is, the larger the $l_2$ distance between two modes can be.

**Theorem 3.3** (NF with $L_2$-Lipschitz inverse mappings $F^{-1}$ fail to capture low density balls). *Let $F^{-1}$ be $L_2$-Lipschitz. We consider the balls centered on $F^{-1}(0)$, we have the lower bound:*

$$\mathcal{D}_{\mathrm{TV}}(P^*, \widehat{P}) \geq \sup_{R} \left( \frac{\gamma(\frac{d}{2}, \frac{R^2}{2L_2^2})}{\Gamma(\frac{d}{2})} - P^*(B_{R, F^{-1}(0)}) \right)$$

*Therefore, if we find a ball for which the the true measure satisfies $P^*(B_{R, F^{-1}(0)}) < \frac{\gamma(d/2, R^2/2L_2^2)}{\Gamma(d/2)}$, then the TV is necessarily strictly positive.*

Both formal proofs are detailed in appendix A.2 and A.3.

### 3.3. Comparison to related work

A related set up is used in (Tanielian et al., 2020). The authors consider two disconnected subsets $M_1$ and $M_2$ separated by a distance $D$, with equal probabilities in the latent

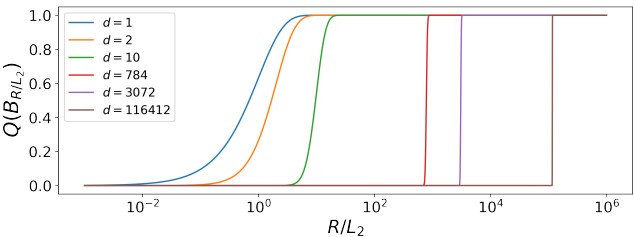

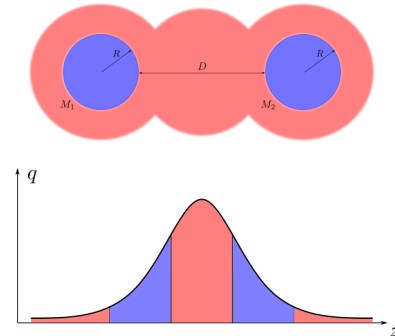

*Figure 3.* Representation of the Gaussian Measure of balls of radius $R/L_2$ centered on 0. The measure is given for dimension 1, 2, 10 and then the dimensions of MNIST (Yann LeCun et al., 2010), CIFAR10 (Alex Krizhevsky, 2009) and CelebA (Liu et al., 2015)

*Figure 4.* Experimental set up given by (Tanielian et al., 2020)

space, i.e. $\widehat{P}(M_1) = \widehat{P}(M_1) = 1/2$. As a consequence, $F^{-1}(0)$ is equidistant from $M_1$ and $M_2$ as illustrated in Figure 4.

**Corollary 3.3.1** (NF with $L_2$-Lipschitz inverse mapping). *If $F^{-1}$ is $L_2$-Lipschitz, then we have a lower bound on the TV distance based on the distance $D$ between $M_1$ and $M_2$:*

$$\mathcal{D}_{\text{TV}}(P^*, \widehat{P}) \geq \gamma(\frac{d}{2}, \frac{D^2}{2L_2^2})/\Gamma(\frac{d}{2}).$$

Note that, the TV distance being defined as the sup on any subspace $A$, we can accumulate the failures made by the network, and therefore take into account the error when the two manifolds $M_1$ and $M_2$ are too dense. This set up is then a appropriate pathological case to study the effect of the bi-Lipschitzness of the mapping.

The original work assesses the learning abilities of their generative model, a GAN (Goodfellow et al., 2014), with a definition of precision and recall given by (Sajjadi et al., 2018) and improved by (Kynkäänniemi et al., 2019). The main advantage of this metric is that is well fitted to be used with the Gaussian Isoperimetric Inequality and therefore gives a result independent from the dimension. By using, the TV distance or any distance for that matter, it can be applied on distributions with any support. The details of the Precision and Recall and the comparison between both methods can be found in appendix B and the proof of corollary 3.3.1 is in appendix A.4.

## 4. Potential remedies & Discussion

As mentioned earlier, increasing the Lipschitz constants of the entire network (for example, by adding extra layers) may impact invertibility and stability during training (Behrmann et al., 2021), and thus is not a suitable approach to improve the expressivity.

Alternatively, one can consider learning the parameters of the latent Gaussian distribution $\boldsymbol{\mu}$ and $\Sigma = \text{diag}(\sigma_i)$. However, this is equivalent to changing the Lipschitz constants

of $F$ from $(L_1, L_2)$ to $(\frac{L_1}{\sigma}, L_2\sigma)$, thus this results in trading off the expected error on very dense subsets (Theorem 3.2) with the expected error on subsets with low densities (Theorems 3.3) or vice-versa. In other words this can lead to a better approximation for a some particular data distributions, but it does not generally improve the expressivity of the normalizing flow.

To improve expressivity, a Gaussian Mixture latent distribution can be considered. Indeed, Khayatkhoei et al. (2019) and Izmailov et al. (2019) have shown that such distributions can learn disconnected manifolds. When the latent distribution is a Gaussian Mixture, Theorem 3.3 does not hold anymore. Limitations similar to the ones highlighted in Theorem 3.2 still apply, but can be mitigated using learnable parameters.

We can trivially adapt the lower bound from Theorem 3.2 to the Gaussian Mixture with K equally distributed modes with learnable mean $\boldsymbol{\mu}_j$ and diagonal covariance matrix $\Sigma_j = \text{diag}(\sigma_{ji})$:

$$\mathcal{D}_{\text{TV}}(P^*, \widehat{P}) \geq \sup_{R, \boldsymbol{x}_0} \left( P^*(B_{R, \boldsymbol{x}_0}) - \frac{1}{K} \frac{RL_1}{\prod_i \sigma_{ji}^d \sqrt{\pi}} \right)$$

As we can see here, the lower bound depends on the inverse of the number of modes $K$ in the mixture. Thus, this approach can solve the limitations highlighted in Theorem 3.2 only if $K$ is small enough (so that the probability mapped onto a dense subset can be sufficiently large). In addition, further investigation is required to understand how to tune or learn the parameters $\sigma_{ij}$ and $K$ of the Gaussian Mixture in order to obtain satisfying training results.

## 5. Conclusion

We have established the bi-Lipschitz constraints reduce the expressivity of Normalizing flows. When the dataset meets some particular conditions such as a high density zone or

a low density zone between two high density zones, the reduced expressivity fails to capture the real distribution of the dataset. To compensate, this lack of learning ability of the mapping, a more complex, i.e. expressive latent distribution can be implemented. However, this method suffers from training difficulties and should be further studied.

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

# A. Proofs

## A.1. Proof of theorem 3.1

By definition we have $\widehat{P}(A) = \int_A \hat{p}(\boldsymbol{x})d\boldsymbol{x}$, then with the change of variable formula we obtain:

$$
\begin{aligned}
\widehat{P}(A) &= \int_A |\mathrm{Jac}_F(\boldsymbol{x})|q(F(\boldsymbol{x}))d\boldsymbol{x} \\
&= \frac{1}{(2\pi)^{d/2}} \int_A |\mathrm{Jac}_F(\boldsymbol{x})|e^{-\|F(\boldsymbol{x})\|^2/2}d\boldsymbol{x}
\end{aligned}
$$

As $F^{-1}$ is $L_2$-Lipschitz, $F$ satisfies:

$$
\forall \boldsymbol{x}_1, \boldsymbol{x}_2 \in \mathcal{X}, \quad \frac{1}{L_2}\|\boldsymbol{x}_1 - \boldsymbol{x}_2\| \leq \|F(\boldsymbol{x}_1) - F(\boldsymbol{x}_2)\|,
$$

and in particular, we have:

$$
\forall \boldsymbol{x} \in \mathcal{X}, \quad \frac{1}{L_2}\|\boldsymbol{x}_1 - F^{-1}(0)\| \leq \|F(\boldsymbol{x})\|.
$$

Consequently $\forall \boldsymbol{x} \in \mathcal{X}$

$$
\begin{aligned}
q(F(\boldsymbol{x})) &= \frac{1}{(2\pi)^{d/2}}e^{-\|F(\boldsymbol{x})\|^2/2}, \\
&\leq \frac{1}{(2\pi)^{d/2}}e^{-\|\boldsymbol{x}/L_2 - F^{-1}(0)/L_2\|^2/2}, \\
&\leq \frac{1}{(2\pi)^{d/2}}e^{-\|T(\boldsymbol{x})\|^2},
\end{aligned}
$$

where $T$ is the affine mapping given by

$$
T(\boldsymbol{x}) = \frac{\boldsymbol{x} - F^{-1}(0)}{4L_2}.
$$

As $F$ is $L_1$-Lipschitz we have $|\mathrm{Jac}_F(\boldsymbol{x})| < L_1^d$ and thus

$$
\begin{aligned}
\widehat{P}(A) &\leq \left(\frac{L_1}{\sqrt{2\pi}}\right)^d \int_A e^{-\|T(\boldsymbol{x})\|_2^2}d\boldsymbol{x} \\
&\leq \left(\frac{L_1}{\sqrt{2\pi}}\right)^d \int_{T(A)} \frac{1}{(4L_2)^d}d\boldsymbol{x} \\
&\leq \left(\frac{L_1}{4L_2\sqrt{2\pi}}\right)^d \mathrm{vol}(A),
\end{aligned}
$$

and thus $TV(P^*, \widehat{P}) = \sup_A |P^*(A) - \widehat{P}(A)|$ implies

$$
TV(P^*, \widehat{P}) \geq
$$
$$
\sup_A \left( P^*(A) - \left(\frac{L_1}{4L_2\sqrt{2\pi}}\right)^d \mathrm{vol}(A) \right)
$$

## A.2. Proof of theorem 3.2

By definition of the TV distance, we have

$$
\mathcal{D}_{\mathrm{TV}}(P^*, \widehat{P}) \geq \sup_{R, \boldsymbol{x}_0} |P^*(B_{R,\boldsymbol{x}_0}) - Q(F(B_{R,\boldsymbol{x}_0}))|,
$$

where $B_{R,\boldsymbol{x}_0}$ is the ball of a radius $R$ centered in $\boldsymbol{x}_0$.

Then, the idea is to show that the image of a ball $B_R$ by a $L_1$-Lipschitz function is in a ball of radius $L_1 R$, and then use a reverse isoperimetric inequality the find an upper bound of the measure of a ball of a radius $L_1 R$.

**Proof of** $F(B_{R,\boldsymbol{x}_0}) \subset B_{L_1 R, F(\boldsymbol{x}_0)}$

First of all, for every $\boldsymbol{z} \in F(B_{R,\boldsymbol{x}_0})$, there exist $\boldsymbol{x} \in B_R$ such that $F^{-1}(\boldsymbol{z}) = \boldsymbol{x}$, we have:

$$
\begin{aligned}
\|F(F^{-1}(\boldsymbol{z})) - F(\boldsymbol{x}_0)\| &= \|F(\boldsymbol{x}) - F(\boldsymbol{x}_0)\| \\
&\leq L_1\|\boldsymbol{x} - \boldsymbol{x}_0\| \\
&\leq L_1 R
\end{aligned}
$$

**Upper bound of** $Q(B_{L_1 R})$  First of all, it can be easily establish that $Q(B_{L_1 R}(F(\boldsymbol{x}_0)))$ is at a maximum when $F(\boldsymbol{x}_0) = 0$. From now on, we will only consider $B_{L_1 R}$ the ball centered on $0$. Therefore the objective is to find an upper bound on:

$$
\begin{aligned}
Q(B_{L_1 R}) &= \int_{\|\boldsymbol{z}\|<L_1 R} q(\boldsymbol{z})d\boldsymbol{z} \\
&= \int_{\|\boldsymbol{z}\|<L_1 R} \frac{1}{(\sqrt{2\pi})^d}e^{-\|\boldsymbol{z}\|^2/2}d\boldsymbol{z}
\end{aligned}
$$

We can use the polar coordinates system to get another expression of the Gaussian measure with $S_{d-1}(r) = \frac{2\pi^{d/2}r^{d-1}}{Q(d/2)}$ being the volume of the hypersphere:

$$
\begin{aligned}
Q(B_{L_1 R}) &= \frac{1}{(2\pi)^{d/2}} \int_0^{L_1 R} S_{d-1}(r)e^{-r^2/2}dr \\
&= \frac{2}{2^{d/2}\Gamma(d/2)} \int_0^{L_1 R} r^{d-1}e^{-r^2/2}dr
\end{aligned}
$$

However $r^{d-1}e^{-r^2/2}$ has a maximum value reached for $r = \sqrt{d-1}$, we can have an upper bound:

$$
\begin{aligned}
Q(B_{L_1 R}) &\leq \frac{2}{2^{d/2}\Gamma(d/2)}\sqrt{d-1}^{d-1}e^{-\frac{d-1}{2}} \int_0^{L_1 R} dr \\
&\leq \frac{\sqrt{2}L_1 R}{\Gamma(d/2)}\left(\frac{d-1}{2e}\right)^{\frac{d-1}{2}}
\end{aligned}
$$

Then, with the Stirling approximation of the Gamma function:

$$
\begin{aligned}
\frac{1}{2}\Gamma(d/2) &= \frac{1}{d}\Gamma(d/2 + 1) \\
&\geq \frac{\sqrt{\pi}\sqrt{d}}{d}(d/2)^{d/2}e^{-d/2} \\
&\geq \frac{\sqrt{\pi}}{2^{d/2}}d^{\frac{d-1}{2}}e^{-\frac{d}{2}}
\end{aligned}
$$

We obtain:

$$
\begin{aligned}
Q(B_{L_1 R}) &\leq \frac{2}{2^{d/2}\Gamma(d/2)}(d-1)^{\frac{d-1}{2}}e^{-\frac{d-1}{2}} \\
&\leq \frac{L_1 R\sqrt{e}}{\sqrt{\pi}}\left(\frac{d-1}{d}\right)^{\frac{d-1}{2}}
\end{aligned}
$$

Using the bound

$$
\frac{1}{\sqrt{e}} < \left(\frac{d-1}{d}\right)^{\frac{d-1}{2}},
$$

we have

$$
Q(B_{L_1 R}) < \frac{L_1 R}{\sqrt{\pi}}
$$

Otherwise, an inequality from (Ball, 1993) give for the hypersphere in $\mathbb{R}^{d-1}$, an upper bound of the Gaussian Measure over a convex set C in $\mathbb{R}^d$:

$$\forall d \geq 2, \int_{\partial C} q < 4d^{1/4}$$

Therefore, for a Ball $B_{L_1 R}$:

$$\forall d \geq 2, \quad Q(B_{L_1 R}) < 4d^{1/4} L_1 R$$

**Lower Bound of the TV**   As soon as we have an upper bound on $Q(B_{L_1 R})$, we have:

$$\mathcal{D}_{\text{TV}}(P^*, \widehat{P}) \geq \sup_{R, \boldsymbol{x}_0} \left( P^*(B_{R, \boldsymbol{x}_0}) - \frac{L_1 R}{\sqrt{\pi}} \right)$$

With the second upper bound, the theorem can be formulated as:

$$\mathcal{D}_{\text{TV}}(\mu^*, \mu_\theta) > \sup_{R, \boldsymbol{x}_0} \left( \mu^*(B_{R, \boldsymbol{x}_0}) - 4d^{1/4} R L_1 \right)$$

### A.3. Proof of theorem 3.3

In this section, we denote $B_R = B_{R, F^{-1}(0)}$. As $F^{-1}$ is $L_2$-Lipschitz, $F^{-1}(B_{R/L_2, 0}) \subset B_R$ and thus

$$\widehat{P}(B_R) \geq \widehat{P}(F^{-1}(B_R)) = Q(B_{R/L_2, 0}).$$

By construction

$$Q(B_{R/L_2, 0}) = \mathbb{P}\left( \|\boldsymbol{z}\|^2 \leq \frac{R^2}{L_2^2} \right),$$

when $\boldsymbol{z}$ follows the standard Gaussian distribution in $\mathbb{R}^d$. This quantity can be computed using the cumulative distribution function of the chi-square distribution, i.e.

$$Q(B_{R/L_2, 0}) = \frac{\gamma(\frac{d}{2}, \frac{R^2}{2L_2^2})}{\Gamma(\frac{d}{2})},$$

where $\gamma$ is the lower incomplete gamma function given by

$$\gamma(x, k) = \int_0^x t^{k-1} e^{-t} dt.$$

### A.4. Proof of Corollary 3.3.1

Since $M_1$ and $M_2$ are separated by a distance $D$ the ball centered on $F^{-1}(0)$ has a radius at least as big as $D$ that we might call $B_D$ to simplify the notation. Therefore:

$$
\begin{aligned}
\bar{\alpha} &= \widehat{P}(M_1) + \widehat{P}(M_2) \\
&= 1 - \widehat{P}(\overline{M_2 \cup M_1}) \\
&\leq 1 - \widehat{P}(B_D) \\
&\leq 1 - Q(F(B_D)) \\
&\leq 1 - Q(B_{D/L_2)}) \\
&\leq 1 - \frac{\gamma(\frac{d}{2}, \frac{D^2}{2L_2^2})}{\Gamma(\frac{d}{2})}
\end{aligned}
$$

And since $P^*(B_D) = 0$:

$$
\begin{aligned}
\mathcal{D}_{\text{TV}}(P^*, \widehat{P}) &\geq |\widehat{P}(B_D) - P^*(B_D)| \\
&\geq \widehat{P}(B_D(F^{-1}(0)) \\
&\geq \frac{\gamma(\frac{d}{2}, \frac{D^2}{2L_2^2})}{\Gamma(\frac{d}{2})}
\end{aligned}
$$

## B. The link with the Precision and Recall for generative models

### B.1. Definitions of the precision and the recall

The precision and the recall are defined as such :

**Definition B.1.** *For $\alpha, \beta \in [0, 1]$, the distributions $\widehat{P}$ is said to have a precision $\alpha$ at recall $\beta$ with respect to $P^*$ if there exist the distributions $\nu, \hat{\nu}, \nu^*$, such that $\widehat{P}$ and $P^*$ can be decomposed as such :*

$$\widehat{P} = \alpha\nu + (1 - \alpha)\hat{\nu} \quad and \quad P^* = \beta\nu + (1 - \beta)\nu^*$$

*The distribution $\nu$ defined on $\text{Supp}(\widehat{P}) \cup \text{Supp}(P^*)$ while $\text{Supp}(\hat{\nu}) = \text{Supp}(\widehat{P})$ and $\text{Supp}(P^*) = \text{Supp}(\nu^*)$*

It can be interpreted as such in (Sajjadi et al., 2018): $\nu$ represent the part of $P^*$ that $\widehat{P}$ correctly models, $\hat{\nu}$ is simultaneously the part of $P^*$ that $\widehat{P}$ misses on their joint support and all the points that should not be represented by $\widehat{P}$. Finally, $\nu^*$ cover the points of $P^*$ that $\widehat{P}$ could not model and the difference between $\nu$ and $\hat{\nu}$ on their joint support. Among all the potential decompositions, i.e. the pairs $(\alpha, \beta)$, the focus is set on the maximum precision $\bar{\alpha}$ and the maximum recall $\bar{\beta}$.

**Proposition B.1.** *The maximum precision and the maximum recall satisfy :*

$$\bar{\alpha} = \widehat{P}(\text{Supp}(P^*)) \quad and \quad \bar{\beta} = P^*(\text{Supp}(\widehat{P}))$$

An improved version of the precision and the recall are defined in (Kynkäänniemi et al., 2019).

### B.2. The link between the maximum precision, the maximum recall and the TV distance

The link between the TV and the maximum precision $\bar{\alpha}$ and the maximum recall $\bar{\beta}$ is :

$$\mathcal{D}_{\text{TV}}(P^*, \widehat{P}) \geq |P^*(\text{Supp}(P^*)) - \widehat{P}(\text{Supp}(P^*))| = 1 - \bar{\alpha}$$

$$\mathcal{D}_{\text{TV}}(P^*, \widehat{P}) \geq |P^*(\text{Supp}(\widehat{P})) - \widehat{P}(\text{Supp}(\widehat{P}))| = 1 - \bar{\beta}$$

### B.3. Corollary 3.3.1 given in terms of maximum precision

**Corollary B.0.1** ($F^{-1}$ $L_2$-Lipschitz)**.** *If $F^{-1}$ is $L_2$-Lipschitz, therefore we have an upper bound on the maximum precision based on the distance $D$ between $M_1$ and*

$M_2$:

$$\bar{\alpha} \leq 1 - + \frac{\gamma(\frac{d}{2}, \frac{D^2}{2L_2^2})}{\Gamma(\frac{d}{2})}$$

This result is to be compared with the actual upper bound on the maximum precision:

$$\bar{\alpha} + \frac{2D}{L_2} e^{-\Phi^{-1}(\bar{\alpha}/2)^2} \leq 1 \text{ where } \Phi(t) = \int_{-\infty}^{t} \frac{\exp(-r^2/2)}{2\pi} dr$$

Our result may depend on the dimension but an upper bound of the precision can be directly computed.