# OpenReview forum: "On the expressivity of bi-Lipschitz normalizing flows"
_ICML.cc/2021/Workshop/INNF — INNF+ 2021 poster_

### Official Review · Reviewer_wjoa · 2021-06-11

**Rating:** Accept
**Confidence:** 3

**Summary:**

The authors prove several lower bounds on the total variation divergence of normalizing flows that are bi-Lipschitz, i.e. have an upper and lower bound on their Jacobian determinant.

**Justification For Rating:**

While the main idea behind the paper is intuitively obvious---Lipschitz constants prevent a normalizing flow from perfectly approximating an arbitrary probability distribution---the concrete bounds are very interesting.

I appreciate that bounds are shown for both overly large target densities and overly small target densities.

My (minor) criticisms of the work are
- Language: the grammar and structure of the text could use another pass. I felt myself stumble quite frequently while reading.
- It would be nice if the beginning of the paper made clearer that bounds are studied for the specific case of a Gaussian latent distribution. The introduction of the Gaussian density seemed more like an example to me on my first read-through.
- Since the option of a latent mixture of Gaussians is suggested as a solution to overcoming the TV bounds: I would also appreciate a more elaborate discussion of the effect that the choice of latent distribution has. Intuitively, I would expect that a change-of-variables from Gaussian to another measure could be folded into the flow, resulting simply in a change of Lipschitz constants. This would allow quanitifying the benefit of the choice of latent distribution within the framework proposed by the authors. Of course, this latter point is just speculation and I would be curious about the authors' thoughts.

---

### Official Review · Reviewer_Tvc8 · 2021-06-11

**Rating:** Borderline Accept
**Confidence:** 3

**Summary:**

Normalizing flows allow parameterizing complex probability distributions via invertible transformations of simple base distributions. The notion of a *Lipschitz continuity* is of great relevance in the context of normalizing flows. For example, bounding Lipschitz constants allows regularizing residual networks to be invertible, resulting in i-ResNet's (Behrmann et el., 2019). It also makes other types of invertible neural networks less prone to numerical errors. Moreover, many normalizing flow models are bi-Lipschitz by design. As a result studying bi-Lipschitz transformations is a fruitful research direction.

In this paper authors set out to understand what effect designing/regularizing normalizing flows to be bi-Lipschitz has on their expressive power. In particular, authors come up with several lower bounds on the total variation distance (TV distance) between the target distribution and its best normalizing flow approximation. The bounds depend on the bi-Lipschitz constants of the invertible transformation used in the flow, and presence of particularly high-density/low-density regions in the target distribution.

Via these bounds authors demonstrate that keeping the flows bi-Lipschitz limits their ability to model certain distributions. Intuitively, authors turn the statement "a  Lipschitz-continuous model will struggle with modelling densities that require significant space expansion/contraction" into a rigorous one.

Finally, authors discuss potential remedies for this phenomenon, including stacking more transformation layers, which would increase expressivity while having limited effect on Lipschitz constant; or using alternative base distributions to reduce the need for significant space expansion/contraction in the first place.

**Justification For Rating:**

I find the premise of the paper interesting. The work is very theoretical, and the conclusion is not a surprising one, but it's useful to make intuitive statements like "keeping a transformation smooth limits its expressivity" rigorous and mathematical. I liked the Figures 1 and 2, which make the sometimes rather dense arguments much more approachable.

I would've liked authors to motivate the choice of the distance measure to put the bounds on a bit better. Does it have certain desirable properties, or is it simply easier to work with theoretically?

In the background section, I rarely see "invertible normalizing flow" used as a phrase, and it does not make a lot of sense. By definition a normalizing flow defines a probability density function via an *invertible transformation*, plus a base density. We never seek to keep the density function itself (i.e. the normalizing flow) invertible.

While I was generally able to follow the line of reasoning, the exposition could use some work. The writing has a considerable amount of typos, poor phrasing, and inconsistencies, which makes a bad impression on the reader. For example, in titles for Figures 1 and 2 a *density* function ($q$) is evaluated on a set of events, while it's likely authors meant to evaluate the probability *measure* ($Q$) (and do so in the text), plus the typesetting seems to be off for some of the $q(\cdot)$ terms. As another example, the last sentence of the first paragraph in section 4 completely went over my head.

The "Potential remedies" section is also somewhat underwhelming, and I'd encourage the authors to solidify the claims here. The final recommendation is to use the Gaussian Mixture base distribution. While it makes sense why it might help, it raises quite a few questions. How exactly do we set the number of modes $K$, which (as authors argue) "must be reasonably low", yet presumably high enough for complex, multi-modal target distributions? How can we learn a non-differentiable parameter $K$ using gradient descent methods, if that's the intention? How do we initialize the components, and how sensitive the flow is going to be to this initialization? Ideally (perhaps in the next version of paper), at least a proof-of-concept with some numerical experiments would make this recommendation much more solid.

All in all, I like the motivation behind the work and enjoyed seeing the intuitive statements be made mathematical, so I recommend accepting the paper. At the same time, I find the writing and the narrative to be somewhat lacking in the current version, with more details left to be desired when it comes to the final recommendation. I encourage the authors to take another pass on these aspects of the paper.

---

### Official Review · Reviewer_3gkz · 2021-06-12

**Rating:** Accept
**Confidence:** 5

**Summary:**

This paper examines how the constraints on practical implementations of normalizing flows -- the constraints on lipschitz constants of the flow and its inverse -- govern their expressivity.

For a flow to be invertible, a flow and its inverse must have finite Lipschitz constants $L_1$ and $L_2$ respectively. A primary concept in flows is have the determinant of the jacobian of the transform changes the measure of the newly formed density defined by the transform. The authors provide a bound on this determinant via the Lipschitz constants of the flow and its inverse (though there is a typsetting error that needs addressing here!). Even though these constraints can essentially be set free by composing many flow steps, there is a cost to doing regarding invertibility and training stability.

The authors provide two examples of how these constraints manifest for certain subsets of the support of a target distribution being modeled by a flow by considering the total variation distance between the two on this subset. They look at 1. an arbitrary sufficiently concentrated subset of the support and 2. balls of density of radius $R$ centered around some coordinate $\mathbf x_0$. The latter allows them to better specify the limitations on expressivity using the exposition provided for the arbitrary subset. They describe two contexts in which expressivity is limited: high density regimes limited by $L_1$ and low density regions limited by $L_2$ (with respect to transforming from a Gaussian or to one, presumably).

The authors provide suggestions of how to mitigate this problem, but do not explore this experimentally, which is fine because this is a small workshop paper (and they should be useful in furnishing the beginning of ideas, not completely exploring them!). This qualitative discussion is also a little undercooked, and could benefit from more theoretical considerations of how the prior plays into creating circumstances that make it easier to learn a target (when one of this limiting conditions of high or low density apply).

**Justification For Rating:**

This paper should be accepted — it fits the bill as exactly what I think the INNF workshop should be about: a well-framed question that has the beginnings of a good analysis that can spur good discussion / considerations for experimentation and further theoretical analysis.

The paper could be useful in providing a theoretical framework for what have been empirically seen problems with flows and contributes to a growing literature on problems that they may need to overcome (e.g., like recent developments on how to structure flows to maintain invertibility).

The authors should continue their investigation on how one can theoretically motivate choosing the right prior given what can be known about the target density. This problem is likely to be better grounded when knows something explicitly about the target density. In particular, when the density can be evaluated, and you want to use a flow to learn to sample from this known density (i.e. you know something about its structure). This seems more well posed for devising prior-target pairs than from the case where one only learns from data.

---

### Decision · Program_Chairs · 2021-06-14

Accept (poster)